# Terahertz saturable absorbers from liquid phase exfoliation of graphite

Vezio Bianchi[1], Tian Carey[2], Leonardo Viti[1], Lianhe Li[3], Edmund H. Linfield[3], A. Giles Davies[3], Alessandro Tredicucci[4], Duhee Yoon[2], Panagiotis G. Karagiannidis[2], Lucia Lombardi[2], Flavia Tomarchio[2], Andrea C. Ferrari[2], Felice Torrisi[2] & Miriam S. Vitiello[1]

Saturable absorbers (SA) operating at terahertz (THz) frequencies can open new frontiers in the development of passively mode-locked THz micro-sources. Here we report the fabrication of THz SAs by transfer coating and inkjet printing single and few-layer graphene films prepared by liquid phase exfoliation of graphite. Open-aperture $z$-scan measurements with a 3.5 THz quantum cascade laser show a transparency modulation ~80%, almost one order of magnitude larger than that reported to date at THz frequencies. Fourier-transform infrared spectroscopy provides evidence of intraband-controlled absorption bleaching. These results pave the way to the integration of graphene-based SA with electrically pumped THz semiconductor micro-sources, with prospects for applications where excitation of specific transitions on short time scales is essential, such as time-of-flight tomography, coherent manipulation of quantum systems, time-resolved spectroscopy of gases, complex molecules and cold samples and ultra-high speed communications, providing unprecedented compactness and resolution.

[1] NEST, CNR-Istituto Nanoscienze and Scuola Normale Superiore, Piazza San Silvestro 12, Pisa I-56127, Italy. [2] Cambridge Graphene Centre, University of Cambridge, Cambridge CB3 0FA, UK. [3] School of Electronic and Electrical Engineering, University of Leeds, Leeds LS2 9JT, UK. [4] Dipartimento di Fisica, Università di Pisa, Largo Pontecorvo 3, 56127 Pisa, Italy. Correspondence and requests for materials should be addressed to F.T. (email: ft242@cam.ac.uk) or to M.S.V. (email: miriam.vitiello@sns.it).

Ultra-short laser pulses in the terahertz (THz) frequency range have great potential for information and communication technologies[1], security and spectroscopy applications[2], and as tools to probe light–matter interaction phenomena[3] across the engineering, physical and biological sciences. Existing technologies for ultra-short THz pulse generation include the near-infrared ultrafast excitation of photoconductive switches[4] and nonlinear crystals[4], which provide broadband emission albeit with poor spectral bandwidth control and low output powers ($\sim \mu W$)[4], free electron lasers[4] and large table-top optical ultrafast lasers[4], which can provide intense ($1\,MV\,cm^{-1}$ electric field) picosecond-pulses with controlled frequencies, but at the expense of a large footprint which hinders practical applications[5]. Quantum cascade lasers (QCLs) are solid state semiconductor sources with superior performances, in terms of compactness and spectral purity, with respect to any other technology for light generation at THz frequencies[6], but presently do not operate passively in the ultra-short pulse regime. In addition to their Watt-range power[7] and small intrinsic linewidth (100 Hz)[8], their 'bandstructure-by-design'[6] allows one to tailor their frequency, bandwidth and pulse width independently[6]. It would thus be beneficial to develop techniques for generating ultra-short pulses directly from THz QCLs, such as passive mode-locking. This requires the development of appropriate saturable absorbers (SAs).

SAs operating in either transmission or reflection are routinely used to mode-lock lasers[9], enabling a train of short pulses to be derived from continuous-wave operation. The key SA parameters include operation wavelength range[9], dynamic response (recovery time)[9] and saturation fluence (i.e., the pulse energy density required to achieve saturation)[9].

Semiconductors can be used as SAs over a wide frequency range (from visible to mid-infrared, MIR)[10,11]. Their recovery time and saturation fluence can be tuned by altering the growth parameters and/or the device geometry[11]. However, due to the limited tuning range/bandwidth in reflection (tens nm)[9] and the inherent free-carrier absorption losses in transmission[12], they are poorly suited for applications in the 1–10 THz range, hindering the corresponding development of passively mode-locked semiconductor lasers. In the THz range, the photon energy (4–50 meV) is smaller than any semiconductor band-gap and the free-carrier absorption, $\alpha_{fr}$, significantly increases as a function of wavelength, following a square law wavelength-dependence ($\alpha_{fr} \sim \lambda^2$) (ref. 6). Several n-doped semiconductors, such as GaAs, GaP and Ge, have been used as THz SAs, at electric fields of tens of $kVcm^{-1}$ (ref. 12). However, intra-cavity integration in the sub-wavelength ($\sim$10–14 $\mu$m deep) cavities of available THz semiconductor lasers cannot be achieved without a significant increase of intra-cavity losses[13]. Although active mode-locking of THz QCLs has been achieved by exploiting injection seeding[14], phase synchronization[15], and by modulating the QCL driving current with an external RF synthesizer[16], enabling the generation of laser pulses of a few ps duration[14], passive mode-locking has not been achieved, due to the absence of suitable SAs in the THz region. As a further constraint, the inherently strong[6–8] electron–phonon interaction in the QCL polar semiconductor gain medium leads to population inversion relaxation times as short as a few ps (refs 6,14).

Graphene[17,18] and carbon nanotubes[19] are potential SAs for ultrafast lasers. Carbon nanotubes can deliver broadband operation by exploiting a distribution of diameters[19], while broadband operation is intrinsic to graphene[17]. This, along with its ultrafast recovery time[20], low saturation fluence[17], and ease of fabrication[21] and integration[22,23], makes graphene a useful SA. Mode-locked lasers exploiting graphene SAs (GSAs) have been demonstrated at frequencies from the visible to the IR[24–29]. To the best of our knowledge, the only reported THz SA using multi-layer graphene grown on the carbon-face of silicon carbide[30] showed a maximum absorption modulation $\sim$10%, too low to alter the intra-cavity field of existing THz QCLs[30]. Furthermore, its intracavity integration within the 10–14 $\mu$m active region of THz semiconductor sources is hindered by the device architecture/material geometry[6]. It would, therefore, be desirable to develop a THz SA with a much higher absorption modulation, as well as one that can be deposited, as required, on any substrate, including on small (etched) trenches.

Liquid phase exfoliation (LPE) of graphite in a water/surfactant solution[18,31] and organic solvents[31–33] (for example, n-methyl-pyrrolidone; n-dimethylformamide; ortho-dichlorobenzene) is ideally suited for the mass-production of graphene. Graphene from LPE can be mixed or combined in dry or liquid form with a host polymer matrix, and can be used to produce both SAs[18,33] and printable inks[34,35]. Inkjet printing then allows for the deposition of films at low temperatures ($<60\,^{\circ}$C), with 1200 dpi resolution[36]. GSA films operating at 1 $\mu$m have been made by vacuum filtration[26], while polymer composite GSAs have been demonstrated at 875 nm (ref. 24), 1.5 $\mu$m (ref. 23) and 2 $\mu$m (ref. 27). Both inkjet printing and vacuum filtration are promising techniques to fabricate GSAs for QCLs, because they can be used to deposit films on a wide range of materials, ranging from Si wafers to flexible plastic substrates[34,35]. Inkjet printing also enables the deposition of films conforming to the substrate features[34].

To date, water-based inks prepared by LPE of graphite require post-processing to remove the surfactant/polymer[37], while the high boiling point of the resulting inks in n-methylpyrrolidone, n-dimethylformamide and ortho-dichlorobenzene hinders solvent removal by evaporation after the coating/printing process[38]. Surfactants (for example, sodium deoxycholate, sodium cholate, Triton X-100) and solvents can have absorption coefficients larger than a few $cm^{-1}$ in the THz region[39], and may adversely affect the absorption of THz radiation in GSAs[40]. Thus, surfactant-free and low boiling point ($<80\,^{\circ}$C) printable inks must be engineered in order to enable low-temperature ($<100\,^{\circ}$C) solvent removal.

Here, we use LPE of graphite to formulate both a water-based ink and also a surfactant-free, low boiling point, ethanol-based ink, and demonstrate THz saturable absorption from films produced by both vacuum filtration and inkjet printing. Through the combination of open-aperture z-scan experiments, transport analysis of field effect transistors (FETs) embedding these inks and Fourier transform infrared (FTIR) spectroscopy, we demonstrate 80% transparency modulation. This paves the way for the integration of graphene with existing sources to realize ultrafast, mode-locked lasers and passive ultrafast components across the THz frequency range.

## Results

**Materials and characterization techniques**. The viscosity, $\eta$[mPa s], surface tension, $\gamma$[mJ m$^{-2}$] and density, $\rho$[g cm$^{-3}$] influence the jetting of individual drops from a nozzle[41]. A figure of merit, $Z = (\gamma \rho a)^{1/2}/\eta$, can be used to characterize the drop formation[42], and assess the jettability of an ink from a nozzle of diameter $a$ (ref. 41). A range of $2 < Z < 24$ has been identified as generally suitable for inkjet printing[33,42]. Here we use a small 21 $\mu$m nozzle (Fujifilm DMC-11610), and formulate an ink with $Z \sim 12$. Low boiling point inks prepared from LPE of graphite previously exploited a two-solvent formulation[43,44], where the mixture of water–isopropyl alcohol[43] or water-ethanol[44] is tuned to improve the affinity of the solvent to the exfoliated flakes.

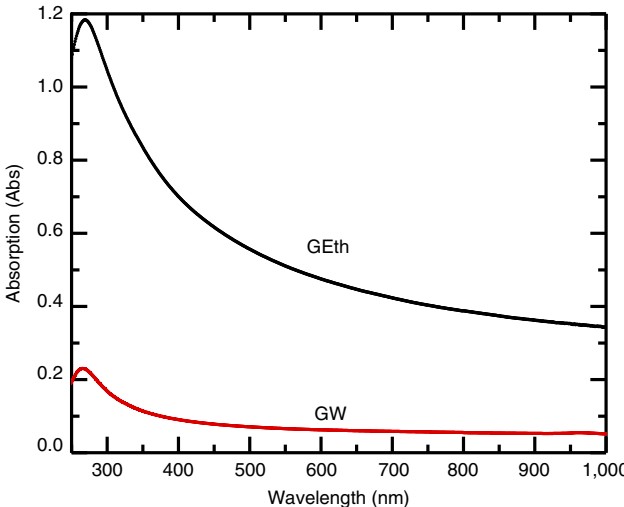

**Figure 1 | Absorption spectra.** Absorption of water-based (GW) and ethanol-based (GEth) inks from the visible to near-infrared range.

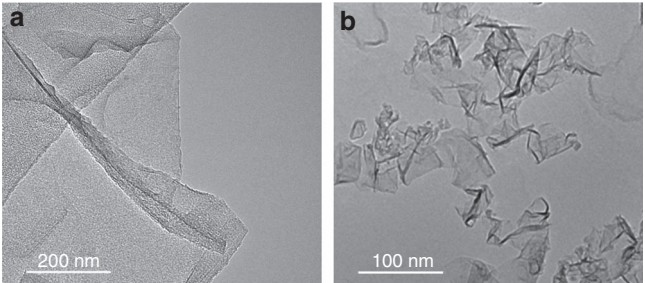

**Figure 2 | Transmission electron microscopy.** Transmission electron microscopy images of (**a**) single-layer graphene flakes from ethanol based (GEth) and (**b**) few-layer graphene flakes from water based inks (GW).

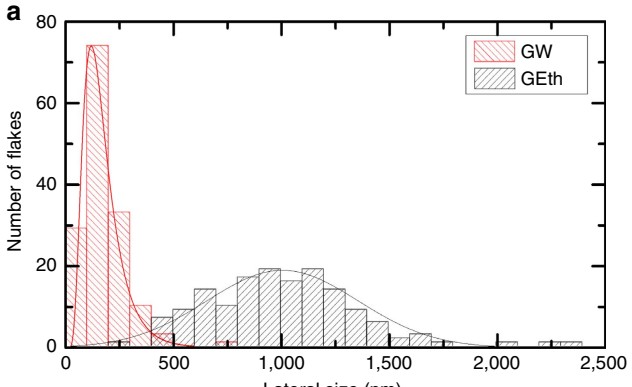

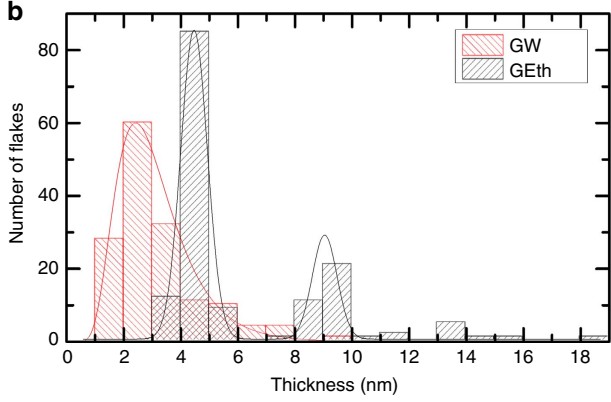

**Figure 3 | Flakes distribution.** (**a**) Flake lateral size distribution and (**b**) thickness, for water-based (GW) and ethanol-based (GEth) inks.

However, the different evaporation rate of the two solvents results in instabilities of $\eta$, $\gamma$ and $\rho$ (ref. 45). Our approach exploits functionalized flakes in low boiling point solvents to formulate stable, surfactant-free, inkjet-printable inks. We print on Si/SiO$_2$ with a roughness $Rq < 0.3$ nm and a water contact angle $\theta_c \sim 62°$. To obtain films on a substrate with uniform flake distribution and morphology, we select a solvent with low contact angle ($<40°$) on SiO$_2$, such as ethanol. We also prepare a water-based ink[31] for vacuum filtration.

The water-based ink (GW) is prepared by ultrasonicating (Fisherbrand FB15069, Max power 800W) the graphite flakes (Sigma Aldrich Graphite) for 9 h in deionized water with sodium deoxycholate (SDC, 9 mg ml$^{-1}$)[31]. The dispersion is then ultracentrifuged (Sorvall WX100 mounting a TH-641 swinging bucket rotor) at 10 k r.p.m. for 1 h to remove thick ($>10$ nm) flakes. After ultracentrifugation, the top 70% of the dispersion is extracted and used for vacuum filtration and film transfer. We also prepare ethanol-based inks (GEth) by ultrasonicating (1 h) 5 mg ml$^{-1}$ flakes (Cambridge Nanosystems, GR1) produced by cracking methane and carbon dioxide in a plasma torch. The dispersion is then ultracentrifuged (Beckman Coulter Proteomelab XL-A, with a SW 32 Ti swinging bucket rotor) at 10 k r.p.m. for 1 h and the top 70% is collected for further characterization. Rheological measurements (viscosity, surface tension, density) give $\eta_{GW} \sim 1.3$ mPa s, $\gamma_{GW} \sim 48$ mN m$^{-1}$, $\rho_{GW} \sim 0.8$ g cm$^{-3}$; $\eta_{GEth} \sim 2.2$ mPa s, $\gamma_{GEth} \sim 30.7$ mN m$^{-1}$, $\rho_{GEth} \sim 0.98$ g cm$^{-3}$, consistent with previous reports[33–38]. Optical absorption spectroscopy is used to estimate the flakes concentration[18,31,32] via the Beer–Lambert law, which correlates the absorbance $A = \alpha cl$, with the beam path length $l$ [m], the concentration $c$ [g l$^{-1}$] and the absorption coefficient $\alpha$ [l g$^{-1}$ m$^{-1}$]. Figure 1 plots the absorption spectrum of GW and GEth inks diluted to 1:20 with water and ethanol respectively, to avoid possible scattering losses at higher concentrations. Using $\alpha \sim 1.390$ l g$^{-1}$ m$^{-1}$ (ref. 31) and $\alpha \sim 2.460$ l g$^{-1}$ m$^{-1}$ (ref. 18) at 660 nm for GW and GEth, respectively, we obtain $c_{GW} \sim 0.1$ mg ml$^{-1}$ and $c_{GEth} \sim 0.36$ mg ml$^{-1}$.

Figures 2a,b show high-resolution transmission electron microscopy (HRTEM) micrographs of a single-layer (SLG) and a few-layer graphene (FLG) flake from the GEth and GW inks, respectively. The associated HRTEM statistics[31] from the GW ink shows $\sim26\%$ SLG, $\sim22\%$ bi- and $\sim18\%$ tri-layers with 150-300 nm average size. HRTEM statistics on the

GEth flakes show $\sim12\%$ SLG, $\sim30\%$ bi- and $\sim58\%$ multi-layers, with $\sim1$ μm average size.

The average lateral size and thickness of the GW and GEth flakes are also estimated by atomic force microscopy (AFM). AFM statistics on the lateral size (Fig. 3a) show a log-normal distribution for the GW flakes peaked at 120 nm with a mean size $\sim150$ nm and a Gaussian distribution for GEth with a mean size $\sim1.04$ μm. The mean GW flake thickness is 2.8 nm (peaked at 2.5 nm) indicating that these are FLGs ($<5$ layers), while $\sim57\%$ of the GEth flakes are 4–5 nm thick (Fig. 3b) and $\sim20\%$ have higher thickness $\sim9$ nm.

GSAs are prepared as follows. The first, W-GSA, is made with the GW ink. Approximately 250 μl is vacuum filtered using 100 nm pore-size nitrocellulose filters. This blocks the flakes,

while water passes through, leading to a film on the surface of the filter. This is then placed on an intrinsic Si/SiO$_2$ wafer and annealed at $\sim 80\,°C$ for 2 h, to improve adhesion, followed by dissolution of the filter in acetone overnight. The resulting film is $\sim 65$ nm thick, as determined by AFM. Following the procedure reported in ref. 27, we estimate a density $\sim 0.72\,g\,cm^{-3}$ for the W-GSA, derived from the weight (measured with a microbalance) of the filter before and after film deposition. This is $\sim 1/3$ of the density of bulk graphite. Since the graphene flakes[27] in W-GSA films are, on average, stacked with the direction parallel to the SiO$_2$ surface, we can assume an interflake distance $\sim 3$ times larger than that of graphite (0.33 nm), which results in $\sim 1$ nm; thus the film thickness corresponds to an equivalent number of layers $N \sim 65$.

The second, Eth-GSA, is prepared by inkjet-printing (Fujifilm Dimatix, DMP-2800) the GEth ink on intrinsic Si/SiO$_2$ at an inter-drop spacing of 50 µm for 100 printing passes. The resulting thickness is $\sim 25$ nm. In this case, once the droplets land on the substrate, they spread according to Young's equation[46,47] $\gamma_{SV} - \gamma_{SL} - \gamma_{LV}\cos\theta_c = 0$, where $\gamma_{SV}$ [mJ m$^{-2}$] is the solid–vapour surface energy, $\gamma_{SL}$ is the solid–liquid interfacial tension, and $\gamma_{LV}$ is the liquid–vapour surface tension[48]. When inkjet printing the GW ink, the high surface tension ($\gamma_W \sim 47.8\,mN\,m^{-1}$) results in de-wetting (that is, isolated droplets caused by high $\theta_c$). On the other hand, the lower surface tension of the GEth ink ($\gamma_{GEth} \sim 30.7\,mN\,m^{-1}$) offers a lower $\theta_c$, which causes the droplets to spread and join, enabling morphologically uniform films on Si/SiO$_2$. The boiling point of GEth ($\sim 78\,°C$) and low surface tension ($30.7\,mN\,m^{-1}$) minimize the effect of the cohesion forces responsible for the transport of material inside and around a coffee ring[49,50].

Raman spectroscopy is used to monitor the quality of the flakes at each step of the GSA preparation process for both inks (Fig. 4a,b). Raman spectra are acquired at 457, 514.5 and 633 nm with a Renishaw InVia with a $\times 50$ objective (N.A. = 0.85). The power on the sample is kept below 1 mW to avoid possible thermal damage. About 20 measurements are taken in different positions on each sample for each excitation wavelength. The G peak corresponds to the high frequency $E_{2g}$ phonon at $\Gamma$, while the D peak, due to the breathing modes of six-atom rings, requires a defect for its activation[51–53]. This arises from the transverse optical phonons around the Brillouin zone edge of the K point[51,53], it is activated by a double resonance (DR) process[54], and is dispersive with the energy of excitation owing to a Kohn anomaly at the K point[55]. Double resonance can also occur through an intra-valley process, that is, connecting two points belonging to the same cone around K or K′, which gives rise to the D′ peak. The 2D peak and the 2D′ peak are the second-order resonances of the D and D′ peaks, respectively. 2D and 2D′ are generated from the momentum conservation by two phonons having opposite wave vectors, meaning that for their activation no defects are required, and thus these modes are always active[56]. The 2D peak is a single Lorentzian in SLG, whereas it splits into several components as the number of layers increases, reflecting the evolution of the electronic band structure[56].

In disordered carbon, the position of the G peak, Pos(G) increases when the excitation wavelength $\lambda_L$ decreases from the IR to UV[52]. Furthermore, the dispersion of the G peak, Disp(G) = $\Delta$Pos(G)/$\Delta\lambda_L$ and the full-width-at-half-maximum of the G peak, FWHM(G) both increase with disorder[52]. The ratio of intensity of the D and G peaks, I(D)/I(G), analysed in combination with FWHM(G) and Disp(G), allows one to discriminate between disorder at the edges of the flakes and in the bulk, where a higher I(D)/I(G) implies a higher FWHM(G) and Disp(G). Figures 4a,b show that the film preparation process does not change the structural properties of the flakes

significantly. On the other hand, Figs 4c–f indicate a difference between the GW-based and the GEth-based films, respectively. The GEth shows a more pronounced and wider D peak, as well as a wider G peak. This implies a more defective nature of these flakes. Furthermore, the higher Disp(G) and FWHM(G) indicate that, unlike the GW flakes, the defects are also spread within the flakes themselves. FWHM(2D) in Fig. 4f confirms that the GEth flakes mostly consist of multilayers. The quality of the GW flakes allows us to qualitatively estimate the Fermi energy ($E_F \le 250$ meV) and, correspondingly, the doping ($\le 4 \times 10^{12}\,cm^{-2}$), by combining Pos(G) with the ratio of the 2D and G integrated areas, A(2D)/A(G) (refs 57–59).

**Optical experiments at THz frequencies**. To investigate the THz-induced non-linear absorption properties of GSAs we use the open-aperture $z$-scan technique[30]. The 3.5 THz radiation generated by a QCL (single-plasmon waveguide)[7] is focused at normal incidence, by using two closely positioned convergent lenses with 3 cm focal length $f$ (Fig. 5a).

The samples are placed on a purpose-designed holder and translated along the optical axis ($z$) using a micrometric stage. A pyroelectric detector with a 7 mm$^2$ sensitive area is positioned at a fixed distance from the laser facet, behind the sample holder, to collect the transmitted radiation. The substrate transmittance is also measured and used to normalize the transmittance data. The QCL is operated in pulsed-mode, under different pulsed regimes. For W-GSA, we use a pulse width $\sim 2$ µs and a modulation frequency 100 kHz, corresponding to a 20% duty cycle. For Eth-GSA, we repeat the experiment twice: under identical conditions as for W-GSA (20% duty cycle); and then by reducing the duty cycle, that is, with a pulse width of 1 µs and a modulation frequency 100 kHz, corresponding to 10% duty cycle, to investigate the sample behaviour under shorter pulses. In all cases, a superimposed additional square function modulation at 33 Hz and with 50% duty cycle is added to match the response of the pyroelectric detector.

To determine the QCL beam profile, we perform knife-edge[60] measurements at the focal point (red dots in Fig. 5b). We record the total power with the pyroelectric detector as the knife-edge is translated through the beam. From these experiments we extract the beam spot size, which has a characteristic Gaussian shape (fitting function, dashed line in Fig. 5b).

Taking into account the nearly circular Gaussian ($x$–$y$) spatial profile and the quasi-continuous wave driving regime of the QCL (so that the beam intensity can be time-independent), the dependence of the absorption coefficient on the pump intensity is expressed as:

$$\alpha(I) = \alpha_{NS} + \frac{\alpha_S}{1 + (I(z)/I_S)} \tag{1}$$

where $\alpha_{NS}$ and $\alpha_S$ represent the non-saturable and saturable components of the linear absorption $\alpha_0 = \alpha(I=0) = \alpha_{NS} + \alpha_S$, respectively, $I(z)$ is the beam intensity along the optical axis, and $I_S$ is the saturation intensity (that is, the intensity at which the saturable absorption is reduced by 50%). $I(z)$ can be written as a function of the beam intensity at the focal point $I_0$, and of the Rayleigh length, $z_R$, which expresses the distance, along the propagation direction of a beam, from the waist to the place where the area of the cross section is doubled:

$$I(z) = \frac{I_0}{1 + (z/z_R)^2} \tag{2}$$

From the measurement of the beam profile (Fig. 5b), we extract a radial spot size $w_0 = 95 \pm 6$ µm, corresponding to $z_R \sim 330$ µm. We also measure at the focal point the average values $I_0 \sim 1.3$ W cm$^{-2}$ for W-GSA, and $I_0 \sim 0.7$ W cm$^{-2}$ and

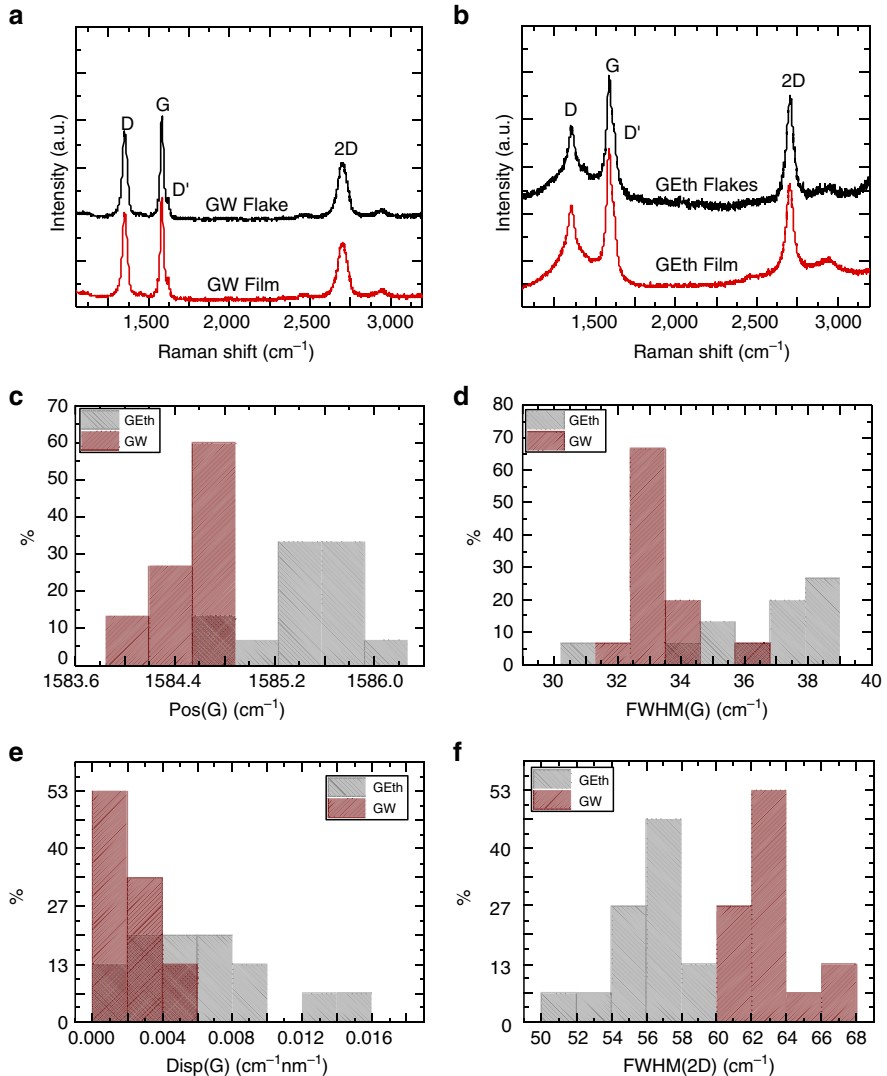

**Figure 4 | Raman spectroscopy.** Raman spectra acquired at 514.5 nm of (**a**) GW flakes (black) and a GW film deposited on SiO₂/Si (red) and of (**b**) GEth flakes (black) and a GEth film (red). Statistics of (**c**) Pos(G), (**d**) FWHM(G), (**e**) Disp(G) and (**f**) FWHM(2D) for GEth (grey) and GW (red) films.

$\sim 1.3\,\mathrm{W\,cm^{-2}}$ for Eth-GSA, due to the different pulsed operating regimes used for Eth-GSA.

FTIR spectroscopy can be used to determine the linear absorption $\alpha_0$ in the THz, MIR and near-IR (NIR) range[61]. Spectra are acquired with deuterated triglycine sulfate (DTGS)-polyethylene, DTGS-KBr and PbSe detectors, optimized for THz, MIR and visible frequencies, respectively[6].

The FTIR transmission spectra of Eth-GSA and W-GSA in the 3–230 THz range, normalized to the SiO₂/Si transmittance, are shown in Fig. 5c,d. These allow us to extract $E_F$, $N$ and to evaluate the carrier scattering dynamics[61].

The optical absorption can be written as a function of the sum of the interband conductivity $\sigma_{inter}$ and the intraband conductivity $\sigma_{intra}$[61]:

$$\sigma_{inter}(\omega)=i\frac{e^2\omega}{\pi}\int_{\Delta}^{\infty}d\epsilon\,\frac{\left(1+\Delta^2/\epsilon^2\right)}{(2\epsilon)^2-(\hbar\omega+i\Gamma_{ib})^2}\times[f(\epsilon-E_F)-f(-\epsilon-E_F)]$$
(3)

$$\sigma_{intra}(\omega)=i\frac{e^2/\pi\hbar^2}{\omega+i/\tau}\int_{\Delta}^{\infty}d\epsilon\left(1+\Delta^2/\epsilon^2\right)\times[f(\epsilon-E_F)+f(\epsilon+E_F)]$$
(4)

where $\omega=2\pi\nu$, $\nu$ is the frequency, $f$ is the Fermi distribution, $\Gamma_{ib}$ describes the broadening of the interband transitions, and $\tau$ is the momentum relaxation time due to carrier intraband scattering. $2\Delta$ represents the material bandgap and is set to zero[61].

The optical transmission through $N$ SLG layers on Si/SiO₂ normalized to the transmission through the reference Si/SiO₂ can be written as[61]:

$$T(\omega)=\left|1+N\sigma(\omega)\sqrt{\mu_0/\varepsilon_0}/(1+n_{sub})\right|^{-2}$$
(5)

where the Si/SiO₂ substrate refractive index, $n_{sub}$, is calculated by using the effective medium theory[62] with a volume ratio (SiO₂ over Si) of $10^{-3}$, and using the values reported in literature for silica glass[63] and silicon in the 1–100 μm range.

By combining equations (3)–(5) and assuming that both interband and intraband processes are present, we get the dependence of the optical transmission on $N$, $E_F$, $\tau$ and $\Gamma_{ib}$ (ref. 61). Eth-GSA (Fig. 5c) shows an almost flat IR transmittance, which allows us to qualitatively evaluate $N\sim 20$ and $E_F \sim 180$ meV via a set of iterative fitting procedures (see Methods and Fig. 6a–c).

Conversely, the visible absorption cut-off in the transmission spectra of W-GSA (Fig. 5d) allows us to retrieve the above

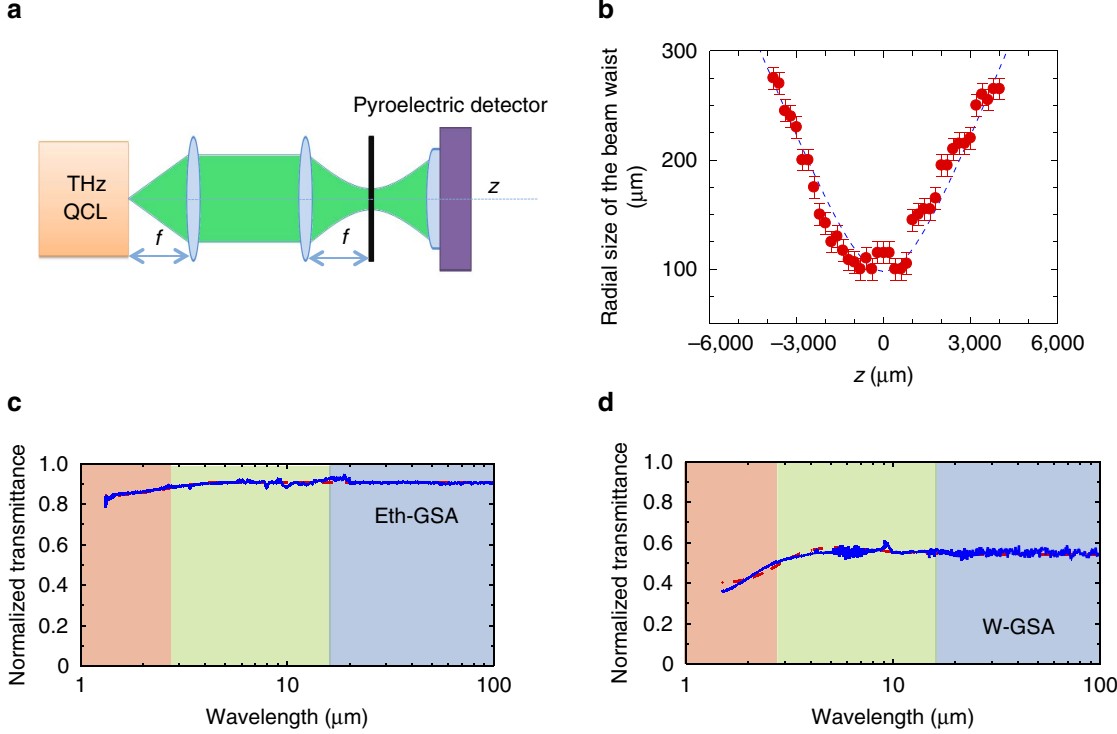

**Figure 5 | Z-scan and FTIR measurements.** (**a**) Schematic diagram showing the open aperture z-scan setup; (**b**) beam waist (radial spot size) profile measured at different positions along the optical axis $z$, of the single-plasmon 3.5 THz QCL operating in pulsed-mode with a pulse width of 2 μs, a modulation frequency of 200 kHz, under an applied field of 7.6 kV cm$^{-1}$. The error bars represent the uncertainty associated with the measurement of the radius of the beam spot. The fitting function $w(z) = w_0 \sqrt{1 + (z^2/z_R^2)}$ allows to retrieve a radial spot size $w_0 \sim 95 \pm 6$ mm. (**c,d**) FTIR transmittance of (**c**) Eth-GSA and (**d**) W-GSA normalized to the transmittance of the SiO$_2$/Si substrate, in the NIR, MIR and THz spectral ranges. The spectra collected separately in the three spectral ranges are plotted together. A white light source is employed in all cases combined with a PbSe detector (NIR) DTGS KBr detector (MIR) and DTGS-polyethylene detector (THz).

parameters via a more detailed procedure. First we estimate the density of free carriers in W-GSA by fabricating a top-gate FET having as active channel the W-GSA film (Fig. 7a) (see Methods). The source-drain distance is set to 100 μm, and the channel width $W_C = 60$ μm, with the top-gate length $L_G \sim 20$ μm (Fig. 7a). We then perform room-temperature transport measurement and investigate the dependence of the source-to-drain current ($I_{SD}$) from the bias applied to the gate electrode ($V_G$), while keeping the source-to-drain bias ($V_{SD}$) equal to 1 mV. The extrapolated sample resistance is $\sim 11$ kΩ at $V_G = 0$ V. By increasing $V_G$, $I_{SD}$ decreases linearly, unveiling a characteristic p-type FET behaviour, meaning that the density of majority carriers (holes) decreases (Fig. 7b). From the linear fit to the data (Fig. 7b, dashed line), we extrapolate the transconductance ($g_m = 4.4$ nA V$^{-1}$) and the expected threshold voltage ($V_{th} = 20.5$ V) for which the linear fit to the data intersects the source-drain current axis. The extrapolated $V_{th}$ allows us to estimate the density of free carriers ($n$) at $V_G = 0$ V, via the relation[64]:

$$n = (C_G \times V_{th})/(e \times A_{gated}) = 4 \times 10^{12} \text{ cm}^{-2} \quad (6)$$

where $C_G = 2.1$ pF is the gate-to-channel capacitance (see Methods), $e$ is the electric charge and $A_{gated}$ is the gated area, calculated assuming that the full transistor channel is influenced by the gate potential. The extrapolated $n$ approximately corresponds to $E_F \sim 250$ meV, which is consistent with the $E_F$ estimated from Raman spectroscopy[56,59]. We then fit the transmission data of Fig. 5d by using equations (3)–(5) and assuming $E_F = 250$ meV as initial value for the iterative fitting procedure. The fitting curve, shown as a dashed line in Fig. 5d,

gives $N \sim 70$ in good agreement with the AFM measurement, $E_F \sim 210$ meV, $\Gamma_{ib} \sim 130$ meV and $\tau \sim 2$ fs.

Figure 8a–c plots the z-scan transmission experiments. We detect an increase in transmission in all samples at $z = 0$, where the z-scan trace is maximum, representing a SA signature. The transmission enhancement is significantly larger (80%) in W-GSA, compared with 2.5% in Eth-GSA. The transmission enhancement is constant (within 5%) in all investigated W-GSA and Eth-GSA samples. On the other hand, the Si/SiO$_2$ substrate shows no changes in transmission along the z-axis, confirming that the absorption bleaching is due to the films.

The normalized transmittance in the z-scan measurements can be written as[65]:

$$T(z) = \left[ 1 - \alpha_0 + \alpha_S - \frac{\alpha_S \left(1 + (z^2/z_R^2)\right)}{1 + (z^2/z_R^2) + (I_0/I_S)} \right] \frac{1}{1 - \alpha_0} \quad (7)$$

By fitting the data in Fig. 8, we extract the saturable and non-saturable absorption coefficients, and the saturation intensity. For W-GSA, $\alpha_{NS} \sim 0.17 \pm 0.02$, $\alpha_S \sim 0.68 \pm 0.1$ and $I_S = 6.7 \pm 1$ W cm$^{-2}$. For Eth-GSA: $\alpha_{NS} \sim 0.08 \pm 0.01$, $\alpha_S \sim 0.17 \pm 0.02$ when the QCL operates at a 10% duty cycle, and $\alpha_{NS} \sim 0.07 \pm 0.01$, $\alpha_S = 0.19 \pm 0.02$ for a 40% duty cycle, showing the negligible role of the device operating conditions. In both cases, the extracted lower saturable absorption values for Eth-GSA reflect the different material quality, in agreement with Raman measurements. The corresponding saturation intensities for Eth-GSA are $I_S = 2.9 \pm 0.7$ W cm$^{-2}$ (10% QCL duty cycle) and $I_S = 3.3 \pm 0.7$ W cm$^{-2}$ (20% QCL duty cycle). In both W-GSA and Eth-GSA the saturation fluence reaches several

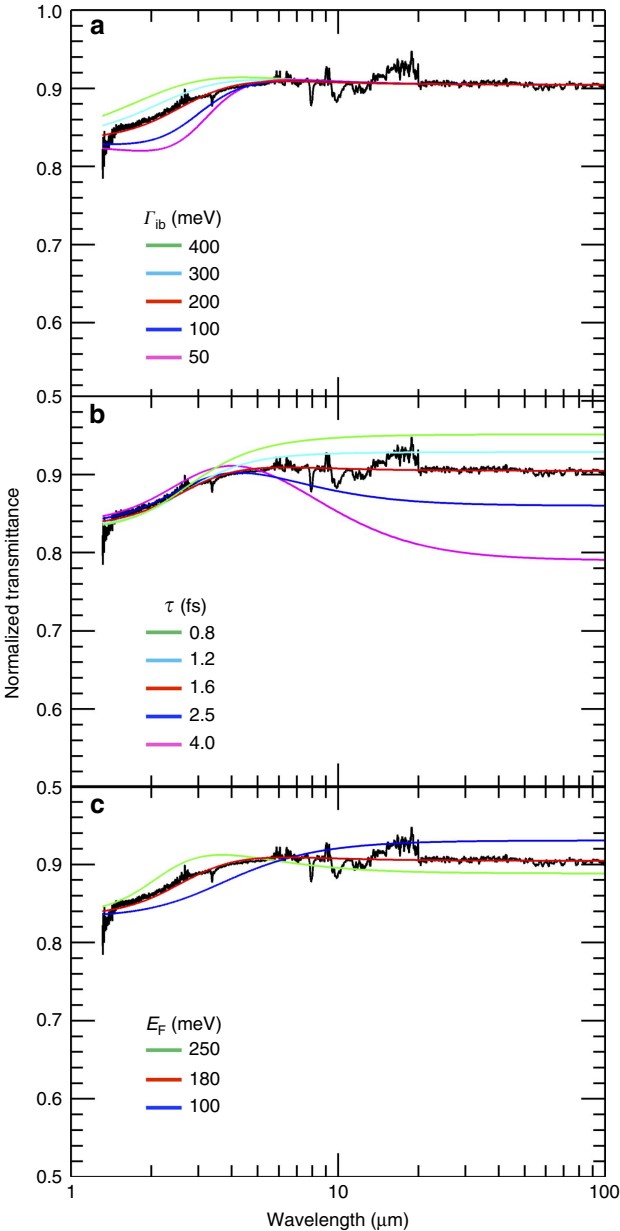

**Figure 6 | Analysis of the FTIR spectra of ethanol-based graphene saturable absorber (Eth-GSA).** (**a**) Dependence on $\Gamma_{ib}$. FTIR transmission spectrum superimposed to fit function ($\Gamma_{ib} = 200$ meV, red line) and a set of functions obtained by varying $\Gamma_{ib}$ in the range 50–400 meV. (**b**) Dependence on $\tau$. FTIR transmission spectrum superimposed to fit function ($\tau = 1.64$ fs, red line) and a set of functions obtained by varying $\tau$ in the range 0.8–4.0 fs. (**c**) Dependence on $E_F$. FTIR transmission spectrum superimposed to the fit function ($E_F = 181$ meV, red line) and a set of functions obtained by varying $E_F$ in the range 100–200 meV.

$\mu$J cm$^{-2}$. The maximum $\alpha_S$ in W-GSA is 68%, while $\alpha_{NS} < 17\%$ on average. The $\alpha_S$ coefficient is the largest reported to date for any spectral range[17,25,28]. The saturation intensity values are in agreement with what achieved in the far-IR[30] in presence of optical pulses much longer (2 $\mu$s) than the demarcation time scale set by the carrier-phonon scattering time ($\sim 150$ fs as derived by pump-probe)[20,66]. Under the above experimental conditions charge carrier distribution develops into quasi-equilibrium within the optical pulse duration.

Typically, non-saturable losses originate from scattering at the interfaces between the layers[67], free carrier absorption[30] or defects,

which depend on the quality of the fabrication techniques. At visible wavelengths, the dominant non-saturable loss is due to scattering losses at the layer interfaces[67], unavoidable even in case of defect-free graphene[67]. Alternative loss dynamics may, however, dominate under specific experimental conditions. For THz excitation, owing to the long wavelength, the multi-interface effect makes only a minor contribution. Additionally, when graphene is doped, non-saturable absorption can be linked to either free carrier absorption or to the reflectivity from the sheets, since in this regime the optical conductivity is well described by the Drude model. Here, at low pumping photon energies (THz frequencies), the optical absorption in graphene is dominated by intraband transitions[68]. Thus, the dynamics of our THz GSA is governed by ultrafast intraband dynamic.

## Discussion

The versatility offered by printable graphene films acting as fast SAs paves the way for new classes of photonic devices in the THz range, with specifically designed device functionalities. The 80% transparency modulation, combined with the flexibility offered by graphene films and inks, can allow, for example, the intra-cavity embedding of SAs in existing THz sources, providing a unique capability for mode-locked and compact THz sources. It will also enable the implementation of ultrafast modulators, filters and mirrors with impacts in the THz frequency range.

## Methods

**Rheological characterization.** The surface tension is measured using the pendent drop method (First Ten Angstroms FTA1000B). The shape of the drop results from the relation between surface tension and gravity. The surface tension is then calculated from the shadow image of a pendant drop using drop shape analysis. The contact angle is measured by dispensing 1 $\mu$l of ultrapure DI water on the substrates and measuring the angle at which the ink interface meets the solid surface. A parallel plate rotational rheometer (DHR rheometer TA instruments (GW ink) and Bohlin C-VOR Rheometer (GEth ink)) is used to evaluate the viscosity as a function of shear rate, the infinite-rate viscosity is found for both the GEth and GW inks.

**Transmission electron microscopy.** Drops of both inks are dispensed on holey carbon TEM grids for HRTEM analysis, using a Tecnai T20 high-resolution electron microscope with an acceleration voltage of 200 kV operating in Bright Field mode.

**Atomic force microscopy.** A Bruker Dimension Icon working in peakforce mode is used. From the centrifuged dispersion four samples are collected and, after 20 times dilution, they are drop cast onto pre-cleaned (with acetone and isopropanol) Si/SiO$_2$ substrate. Each sample is scanned across three different areas. For each material, 150 flakes are counted.

**Vacuum filtration transfer.** $\sim 1$ ml GW ink is diluted with DI water at a ratio of 1:9 and is passed through a nitrocellulose membrane (100 nm pore size), hastened with the use of a Büchner flask attached to a vacuum pump. The filtered film is then washed with DI water in order to remove residual surfactant. The flakes on the membrane are then rinsed in DI water for 5 min and transferred onto an intrinsic Si/SiO$_2$ substrate. After oven annealing ($\sim 80\,°$C) the sample is placed in an acetone bath to dissolve the nitrocellulose membrane.

**Transistor fabrication.** A top-gate FET is fabricated using W-GSA. Source and drain electrodes, shaped as stripes 100 $\mu$m distant from each other, are defined via a combination of electron beam lithography and thermal evaporation of a 5/100 nm sequence of Cr/Au. This geometry defines a channel having a 100 $\mu$m length and an average width of 60 $\mu$m, set by the shape of the selected portion of graphene.

A high-$k$ dielectric (HfO$_2$, $\varepsilon_r = 13$–19)[69] oxide layer is then placed over the whole transistor plane via atomic layer deposition (ALD). In order to guarantee a good electrical isolation a thickness of 100 nm is chosen for this layer.

The gate electrode is designed as a rectangular stripe having a width $W_G = 20\,\mu$m. It is lithographically patterned across the central part of the FET channel with a length (120 $\mu$m) suitable to cover the whole channel. The gate to channel capacitance is with a finite element method (COMSOL Multiphysics). We get $C_G \sim 2.0$–3.0 pF.

**Fourier infrared spectroscopy analysis of Eth-GSA samples.** Figure 6a plots the FTIR transmission spectrum superimposed to the fit achieved by keeping $N$, $E_F$ and $\tau$ as fixed parameters and by varying $\Gamma_{ib}$ in the 50–400 meV range. The fitting curve (red curve in Fig. 6a) that well matches the experimental data corresponds to

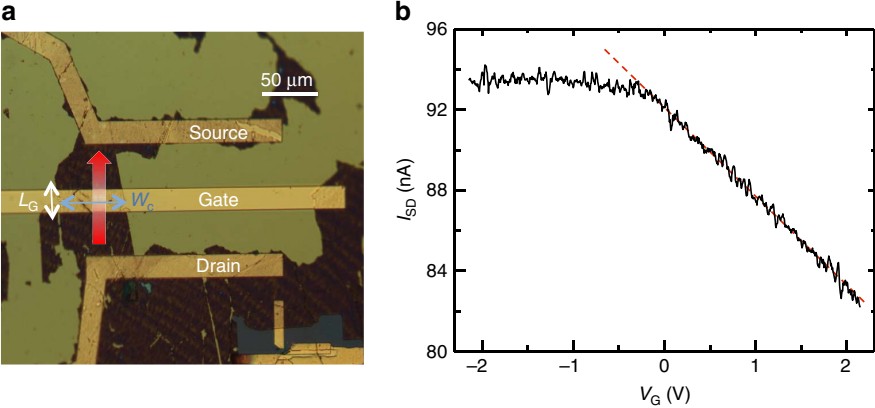

**Figure 7 | Field effect transistor and transport data.** (**a**) Optical microscope image of a top gate FET with a channel made by printing the water-based ink (W-GSA sample). (**b**) Dependence of the source-drain FET current ($I_{DS}$) from the gate bias $V_G$ in the range from $-2.2$ V to $+2.2$ V, measured while a source-drain voltage $V_{SD} = 1$ mV is applied. The dashed line represents the fit to the data in the linear region and gives a transconductance $\sim 4.4$ nA V$^{-1}$.

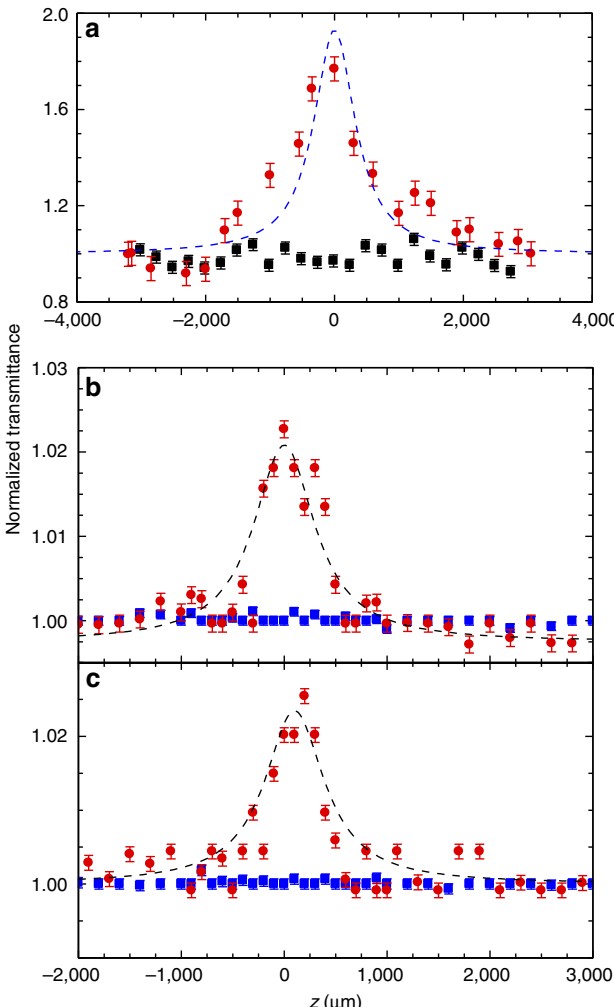

**Figure 8 | Transparency modulation.** (**a**–**c**) z-scan normalized transmittance traces of (**a**) water-based graphene saturable absorber (W-GSA) (dots) and (**b**,**c**) ethanol-based graphene saturable absorber (Eth-GSA) probed with a quantum cascade laser operating at (**b**) 10% duty cycle, and (**c**) 20% duty cycle, compared with the z-scan normalized transmittance traces of the Si/SiO$_2$ substrate (squares). The error bars correspond to the uncertainty interval on the measured normalized transmittance. The dashed lines are the fits with equation (7).

$\Gamma_{ib} = 200$ meV and gives: $N \sim 19$, $E_F \sim 181$ meV, corresponding to $n \sim 1.5 \times 10^{12}$ cm$^{-2}$, $\tau \sim 1.6$ fs. Figure 6b interpolates the data by exploiting the same fit function in which $N$, $E_F$ and $\Gamma_{ib}$ are fixed parameters. $\tau$ is varied in the range 0.8–4.0 fs. The fitting curve (red curve in Fig. 6b) corresponds to $\tau = 1.6$ fs and gives: $N \sim 19$, $E_F \sim 181$ meV and $\Gamma_{ib} \sim 200$ meV. Finally we investigate the influence of $E_F$ on the fit. We keep $N$, $\tau$ and $\Gamma_{ib}$ as fixed parameters and vary $E_F$ in the 100–200 meV range, obtaining: $N \sim 19$, $\Gamma_{ib} \sim 200$ meV and $\tau \sim 1.6$ fs with $E_F = 180$ meV.

**Data availability.** The data that support the findings of this study are available from the corresponding author on request.

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

## Acknowledgements

We acknowledge funding from ERC grants SPRINT, Hetero2D, HiGRAPHINK, the EU Graphene Flagship, and EPSRC grants EP/K01711X/1, EP/K017144/1, EP/L016087/1 and EP/N010345/1.

## Author contributions

M.S.V. conceived the experiments. V.B. and L.V. performed the optical experiments; T.C. and F.T. prepared the graphene inks, performed TEM characterization and printed the graphene saturable absorbers. T.C. performed UV–vis, Rheometry and AFM characterization. T.C., L.V., D.Y., P.G.K., L.Lo., F.T., A.C.F. performed and analysed the micro-Raman experiments; V.B., L.V. and M.S.V. modelled and analysed the data. V.B., L.V., T.C., A.T., A.C.F., F.T., M.S.V. discussed the results. L. Li., E.H.L. and A.G.D. grew the QCL material. M.S.V. fabricated and characterized the QCL. L.V. fabricated and characterized the FET device. M.S.V., F.T. and A.C.F. wrote the manuscript with contributions and discussions from all authors.

## Additional information

**Competing interests:** The authors declare no competing financial interests.

**Publisher's note**: 

