## [Peer Review File · Nature Communications]

Reviewers' Comments:

Reviewer #1:

Remarks to the Author:

Bianchi et al. study the properties of LPE graphite flakes as saturable absorbers in the THz range. The work is nicely carried out, with great attention to detail and all the information needed to reproduce their work is included. There is no doubt this is good work. There is also, of course, great interest in achieving passive modelocked operation in QCL and in the THz. On the other hand, I am not sure the level of advance shown in this manuscript is sufficient to justify publication in Nature Communications.

My main reasons for these are the following:

1. The main (sole?) goal of the saturable absorber is to achieve mode-locked operation. If they had demonstrated mode-locked operation, this would have been a no-brainer, they could probably aim for a Nature Photonics level journal, but they do not show mode-locked operation here. Of course, that will be their next step and I am sure they are working on this already. Ideally, I would like them to demonstrate modelocked operation, at the very least they should discuss openly the remaining challenges that have prevented them from achieving this thus far (for example, is large saturation fluence an issue?, are there undesired dynamics when the GSA is incorporated in the cavity?). In sum, what is currently preventing them from demonstrating mode-locked operation in a QCL with a W-GSA?

2. Ref.30 "Bianco, F. et al. THz saturable absorption in turbostratic multilayer graphene on silicon carbide, Optics Express 23, 11632 (2015)." is a very closely related paper by the authors, the approach and the study are identical. The novelty here is on the fabrication of the GSA and the performance enhancement from 10% to 80% transparency modulation (Fig.8(a)). Looking at the two papers, saturation fluence and ratio of saturable to non-saturable losses are very similar. The improvement in transparency modulation is impressive, but only relevant after they demonstrate mode-locking.

I have a more technical question regarding the Z-scan measurements, this is not to criticize the paper, but if my reasoning is valid, it could make for an interesting experiment. It also should be important in how they build the ML laser cavity (similar to what was done here Zaugg, C. A., et al. " Optics express 21.25 (2013): 31548-31559.):

3. I agree the Z-scan measurement is the best way to measure the non-linear absorption properties of a material. In this case the authors are measuring flakes with a thickness of 3nm (for GW), am I right? The 3.5THz signal has a wavelength of 100um, (10^4 x the flake size). Wouldn't the Z-scan measurements strongly depend on whether you are in a node or an antinode? Is this taken into consideration? If I am not wrong, by changing the gap between the collimating lenses in, say 5um steps, you should see different absorption characteristics. May be the authors have already considered this, but it is not clear in the paper.

Finally, as a suggestion, I feel the authors have made the paper unnecessarily complicated. The only GSA that performs better than [30] is W-GSA. Do we need all the information regarding Eth-GSA? Some people may do, so it can be included in the SI, but in my opinion the main paper only needs the W-GSA information.

All things considered, I would be very happy to accept publication if they demonstrated mode-locked operation. As I suspect they cannot do that, I would like to see some discussion regarding the remaining challenges to achieve their final goal before I can make a recommendation.

Reviewer #2:

Remarks to the Author:

General comment

This paper gives a good overview of the liquid phase exfoliation of graphite, and its application to Terahertz saturable absorbers. The result achieved is at the state of the art in terms of transparency modulation.

In this paper, two graphene inks are used (water-based and ethanol-based ink). Similarly for material deposition, two separate techniques are used: vacuum filtration for water-ink and ink-jet printing approach for ethanol ink. The best result is achieved with water based ink and vacuum filtering deposition. I wonder how the film thickness impacts on the results. It's similar for deposition technique.

Other comments

Comment 1:

The HRTEM image of the GETH flakes (Fig. 2-b) shows of GETH flakes. The dimensions are not consistent with description of lines (138-148)

Comments 2: Lines 194-195

Figs. 5 c,d are related to the comparison of the GW film and GETH film as mentioned in the graph title. But lines 194-195 seem to introduce confusion between flakes and films.

Comments 3:

The procedure describes in the lines 263-279 is quite simplified, and introduces errors such as threshold voltage (V_{th}) concerning graphene field effect transistor (FET). The current (I_{SD}) could not be zero !! This part needs to be change or remove.

Reviewer #3:

Remarks to the Author:

The authors extend results connected with optical properties of graphene/multi-layer graphene films (particularly in the MIR, to the level of a few THz). The manuscript gives wide descriptions of technology of two materials based on the liquid phase exfoliation of graphite:

- so called water-based ink (GW), and
- so called ethanol-based inks (GETh).

Many methods were used to measure the properties and parameters of both materials;

- rheological measurements (viscosity, surface tension, density),
- optical measurements (flake concentrations, absorption coefficients),
- Raman spectroscopy (quality of flakes),
- TEM analysis, etc.

The technology of ink-jet printing is described. Different diagnostic techniques were used for characterization of the flakes, and all necessary and available steps were applied to characterize the flakes .

The aim of this work was to measure the absorption parameters of graphene-based saturable absorbers by using open-aperture z-scan technique.

Firstly, authors measured the normalized transmittance for both materials in the 3 to 230THz (1.3-100 μ m) range, showing that they are quite flat in that range.

Using experiments with graphene based transistor they are able to fit characteristics of transmittance (very clever analysis!).

And the most important results:

using coherent pulsed QCL source at 3.5 THz they made z-scans of both materials. That is real mail-stone in investigations of multi-layer graphene as a saturable absorber. The result is absolutely amazing. They measured the transparency modulation of GW at the level of 80%, which is real breakdown in this area.

Unfortunately for GETh this parameter is much worse – on the level of a couple percent.

The saturation intensities - $I_s = 6.7 \text{ W/cm}^2$ and $I_s = 2.9 \text{ W/cm}^2$ were measured for both materials, respectively.

Summing up, this result is opening the bright future for the MIR fast devices.

I fully recommend it for publications. It will be a seed paper for many new ideas and new investigations.

Reviewer: 1

- **1.** The main (sole?) goal of the saturable absorber is to achieve mode-locked operation. If they had demonstrated mode-locked operation, this would have been a no-brainer, they could probably aim for a Nature Photonics level journal, but they do not show mode-locked operation here. Of course, that will be their next step and I am sure they are working on this already. Ideally, I would like them to demonstrate mode-locked operation, at the very least they should discuss openly the remaining challenges that have prevented them from achieving this thus far (for example, is large saturation fluence an issue?, are there undesired dynamics when the GSA is incorporated in the cavity?). In sum, what is currently preventing them from demonstrating mode-locked operation in a QCL with a W-GSA?

Answer

It is indeed certainly our major goal, and already an on-going research route, to demonstrate mode-locking in lasers operating at THz frequencies exploiting the W-GSA.

The present paper is however not focused on that, since integrating W-GSA, in a proper fashion, for example in a THz semiconductor laser cavity to reach the goal of passive mode-locking requires a totally different device/waveguide architecture and a totally different experimental scheme to probe the latter effect, whose setting is out from the scope of the present manuscript.

Engineering such a novel laser cavity of a state of the art far-infrared miniaturized laser source, and the related mode-locking experimental set-up, certainly deserves a dedicated research manuscript by itself.

The challenging saturable absorber design, fabrication and optical testing that we present in the present manuscript, together with the achieved 80% transparency modulation at Terahertz frequencies, is a major goal by itself.

It is indeed worth underlying that semiconductor saturable-absorbers and saturable-absorber mirrors that are routinely used for mode-locking in the visible and infrared ranges are poorly suitable for applications at THz frequencies for several reasons: i) the photon energy is smaller than the semiconductor band gap; ii) the free carrier absorption induced by the semiconductor doping is a dominant loss source; iii) they require complex fabrication and challenging integration. Although n-doped semiconductors, such as GaAs, GaP and Ge, have been recently used as THz SAs, at tens kV/cm electric fields, intra-cavity integration in the sub-wavelength (~10–14 μm deep) cavities of available THz semiconductor lasers cannot be achieved without a significant increase of intra-cavity losses, which will consequently prevent lasing.

Furthermore, as discussed in the manuscript THz SA using multi-layer graphene grown on the carbon-face of silicon carbide have been demonstrated, but they showed a maximum absorption modulation of ~10% (limited by the huge defect distribution), modulation which is too low to alter the intra-cavity field of existing THz QCLs. Furthermore, its intracavity integration within the 10–14 μm active region of THz QCLs is inherently hindered by the device architecture/material geometry, therefore hindering any possible passive mode locking geometries.

The demonstration of a flexible SA:

- with a significantly higher (80%) absorption modulation;

- that can be deposited, as required, on any substrate, including on small (etched) trenches and *intracavity*;
- at THz frequencies

is a major technological demonstration by itself.

Saying that, our achievement clearly opens the route to the design of completely novel THz semiconductor laser resonator architecture, which can allow, for the first time, the intracavity embedding of a SA.

The beauty of what we demonstrated here is that there are no issues neither with the large saturation fluence nor with possible undesired dynamics when the developed GSA-ink is incorporated in the cavity.

Demonstrate mode-locking means a new geometry and the development of a new laser architecture, and, being absolutely straightforward, this deserves a completely separated study.

Furthermore, mode locking is certainly not the “sole” goal of this study: the versatility offered by printable graphene films acting as fast saturable absorbers paves the way for new classes of photonic devices in the THz range, with specifically designed device functionalities. Our demonstration will also enable the implementation of ultrafast modulators, filters and mirrors with groundbreaking impacts in the Terahertz frequency range.

- **2.** Ref.30 “Bianco, F. et al. THz saturable absorption in turbostratic multilayer graphene on silicon carbide, *Optics Express* 23, 11632 (2015).” is a very closely related paper by the authors, the approach and the study are identical. The novelty here is on the fabrication of the GSA and the performance enhancement from 10% to 80% transparency modulation (Fig.8(a)). Looking at the two papers, saturation fluence and ratio of saturable to non-saturable losses are very similar. The improvement in transparency modulation is impressive, but only relevant after they demonstrate mode-locking.

Answer

Although, as we clearly state in the manuscript it is true that saturable absorption has been preliminary investigated in the THz we do not agree with the statement above.

There are four important differences.

a) The previous THz SA exploiting multi-layer graphene grown on the carbon-face of silicon carbide showed a maximum absorption modulation ~10% (limited by the huge inherent defect distribution), modulation which is too low to alter the intra-cavity field of existing THz quantum cascade laser sources;

b) furthermore and more importantly, the intracavity integration of such epitaxial growth multi-layer THz SA within the 10–14 μm active region of THz QCLs is inherently hindered by the device architecture and the material geometry and flexibility, therefore making impossible to conceive a possible passive mode-locking strategy; passive mode locking is a major, dreamed and unreached goal in the far-infrared.

c) defects limit the saturable absorption and are certainly a major obstacle to any possible combination with THz QCL sources;

d) we here proved that the demonstrated saturable absorbers exploit intraband-controlled absorption bleaching and not interband dynamics, like it was in the previous case, therefore we have a fast absorber, ideal for mode locking.

Therefore, as we discussed in the text, the improvement is major, the architecture idea is different, the demonstrated appealing material properties can allow now to conceive a passive mode locking strategy and this is absolutely straightforward in the Terahertz.

- I have a more technical question regarding the Z-scan measurements, this is not to criticize the paper, but if my reasoning is valid, it could make for an interesting experiment. It also should be important in how they build the ML laser cavity (similar to what was done here Zaugg, C. A., et al. " *Optics express* 21.25 (2013): 31548-31559.):

3. I agree the Z-scan measurement is the best way to measure the non-linear absorption properties of a material. In this case the authors are measuring flakes with a thickness of 3nm (for GW), am I right? The 3.5THz signal has a wavelength of 100um, (10^4 x the flake size). Wouldn't the Z-scan measurements strongly depend on whether you are in a node or and antinode? Is this taken into consideration?

Answer

We thank the referee for his/her comment.

The flake thickness of our graphene saturable absorbers is 60 nm and not 3 nm.

Indeed, in the mentioned paper by *Zaugg et al.* the GSA is placed in front of a DBR mirror which acts as external cavity for the near infrared laser. The tunable displacement between the mirror and the graphene sheet is, in that case, the key parameter that determines the field strength at the graphene sheet position.

However, the configuration proposed by *Zaugg et al.* is based on the interference between a wave that is incident on the mirror and the reflected one; in other words, the saturable absorption tunability is achieved by selecting the position of the graphene sheet with respect to the intensity pattern of a stationary wave.

On the contrary, our z-scan measurements are done in transmission mode, without the presence of a stationary wave along the optical path.

The concept of node and antinode is therefore not straightforwardly defined: the node (or antinode) positions are fixed in space only if a stationary wave pattern exists. Moreover, even in the case where a standing wave is present, the position of the graphene sheet with respect to the nodes/antinodes pattern would not be changed by changing the position along the z axis of the graphene/mirror system. Instead, it would change only if the position of the graphene plane is modified with respect to the position of the mirror (for example while changing the thickness of the SiO₂ layer as in the experiment described by *Zaugg et al.*).

For this reason, the two configurations are very different, we have a saturable absorber and not a “*saturable absorber mirror*”, which is the one mentioned in the cited paper.

The development of a saturable absorber mirror could be also a possible route to consider in the future.

4. If I am not wrong, by changing the gap between the collimating lenses in, say 5um steps, you should see different absorption characteristics. May be the authors have already considered this, but it is not clear in the paper

Answer

the effect suggested by the reviewer can take place only if a strong reflector is placed along the optical path of the THz beam. In our case, the interface of highest reflectance is the silicon/air interface on the back of the substrate where the GSA is deposited. In normal incidence configuration, the reflectance of this interface is less than 30% in the THz range. This value is too low for the formation of a standing wave. Furthermore, as stated above, the meaningful distance in the experiment suggested by the reviewer is the graphene/reflector distance, that, in our geometry, is uniquely determined by the silicon substrate thickness and not by the lenses position.

5. Finally, as a suggestion, I feel the authors have made the paper unnecessarily complicated. The only GSA that performs better than [30] is W-GSA. Do we need all the information regarding Eth-GSA? Some people may do, so it can be included in the SI, but in my opinion the main paper only needs the W-GSA information.

Answer

We thank the reviewer for this comment. It is certainly sure that Eth-GSA does not perform so well like W-GSA. However, despite for the specific target of mode-locking this is not a good option we feel that is certainly of interest for a broader audience of readers to show that we can develop optically active graphene-based devices using industry-level scalable techniques like vacuum filtration (W-GSA) and also inkjet

printing (Eth-GSA), since this is of utmost importance for future implementations of 2D materials for new and cost-effective technological solutions.

Reviewer: 2

This paper gives a good overview of the liquid phase exfoliation of graphite, and its application to Terahertz saturable absorbers. The result achieved is at the state of the art in terms of transparency modulation. In this paper, two graphene inks are used (water-based and ethanol-based ink). Similarly for material deposition, two separate techniques are used: vacuum filtration for water-ink and ink-jet printing approach for ethanol ink. The best result is achieved with water based ink and vacuum filtering deposition. I wonder how the film thickness impacts on the results. It's similar for deposition technique.

Answer

We thank a lot the reviewer for the positive comments and for underlying the state-of-the art results.

The film thickness is expected to affect the saturation intensity and the transparency modulation of the GSAs. In particular, thicker films allow higher saturation intensity and a larger transparency modulation (see ref. [30]). In our case, the different results obtained for W-GSA and Eth-GSA is not solely related to the different film thicknesses. The thickness of W-GSA is approximately twice the thickness of Eth-GSA. This difference certainly here influences the higher saturation intensity of W-GSA ($6.7 \pm 1 \text{ W/cm}^2$, vs. $2.3 \pm 0.7 \text{ W/cm}^2$), and, only partially, the transparency modulation.

The deposition technique impacts on the uniformity of the graphene film. This affects the absorption coefficient of the material, but affects equally the non saturable and saturable contributions. Therefore the dependence from the deposition technique can be described as a scale factor for the $\alpha(z)$ plot.

- **Comment 1:** The HRTEM image of the GE_{th} flakes (Fig. 2-b) shows of GE_{th} flakes. The dimensions are not consistent with description of lines (138-148)

Answer

We thank the referee for his/her comment. The description of the two samples have been mixed up for a trivial mistake. We revised the text has follows:

Line 138

We modified the text:

Figures 2a,b show high-resolution transmission electron microscopy (HRTEM) micrographs of a single-layer (SLG) and a few-layer graphene (FLG) flake from the GW and GE_{th} inks, respectively (Tecnai T20). The associated HRTEM statistics ³¹ from the GW ink shows ~26% SLG, ~22% bi- and ~18% tri-layers, with 150-300 nm average size. HRTEM statistics on the GE_{th} flakes show ~12% SLG, ~30% bi- and ~58% multi-layers with ~1 μm average size.

with the text:

Figures 2a,b show high-resolution transmission electron microscopy (HRTEM) micrographs of a single-layer (SLG) and a few-layer graphene (FLG) flake from the GW and GE_{th} inks, respectively (Tecnai T20). The associated HRTEM

statistics³¹ from the GW ink shows ~12% SLG, ~30% bi- and ~58% multi-layers with ~1 μm average size HRTEM statistics on the GETH flakes show ~26% SLG, ~22% bi- and ~18% tri-layers, with 150-300 nm average size.

Caption Figure 2

modified as follows:

Figure 2. Transmission electron microscopy. Transmission electron microscopy images of (a) SLG flake from GW ink and (b) FLG flakes from GETH ink.

- **Comments 2:** Lines 194-195 Figs. 5 c,d are related to the comparison of the GW film and GETH film as mentioned in the graph title. But lines 194-195 seem to introduce confusion between flakes and films.

Answer

We thank the referee for pointing out the imprecise statement
The text at lines 194-195 has been modified as follows:

On the other hand, Figs. 4c and 4d indicate a clear difference between the GW-based and the GETH-based films, respectively

- **Comments 3:** The procedure describes in the lines 263-279 is quite simplified, and introduces errors such as threshold voltage (V_{th}) concerning graphene field effect transistor (FET). The current (I_{SD}) could not be zero !! This part needs to be change or remove.

Answer

We thank the referee for the comment. The threshold voltage is determined correctly: it is calculated as the voltage at which the linear regression would cross the axis $I_{SD} = 0$.
We modified the text accordingly.

Line 271

We modified the text as follows:

From the linear fit to the data (Fig. 7b), we extrapolate the transconductance ($g_m = 4.4 \text{ nA/V}$) and the expected threshold voltage ($V_{th} = 20.5 \text{ V}$) for which the linear fit to the data (dashed line in Fig. 7b) intersects the source-drain current axis.

Reviewer #3

The authors extend results connected with optical properties of graphene/multi-layer graphene films (particularly in the MIR, to the level of a few THz). The manuscript gives wide descriptions of technology of two materials based on the liquid phase exfoliation of graphite: - so called water-based ink (GW), and - so called ethanol-based inks (GETH).

Many methods were used to measure the properties and parameters of both materials;
- rheological measurements (viscosity, surface tension, density);
- optical measurements (flake concentrations, absorption coefficients),
- Raman spectroscopy (quality of flakes);

- TEM analysis, etc.

The technology of ink-jet printing is described. Different diagnostic techniques were used for characterization of the flakes, and all necessary and available steps were applied to characterize the flakes. The aim of this work was to measure the absorption parameters of graphene-based saturable absorbers by using open-aperture z-scan technique. Firstly, authors measured the normalized transmittance for both materials in the 3 to 230THz (1.3-100 μ m) range, showing that they are quite flat in that range.

Using experiments with graphene based transistor they are able to fit characteristics of transmittance (very clever analysis!). And the most important results:

using coherent pulsed QCL source at 3.5 THz they made z-scans of both materials. That is real milestone in investigations of multi-layer graphene as a saturable absorber. The result is absolutely amazing. They measured the transparency modulation of GW at the level of 80%, which is real breakdown in this area. Unfortunately for GETH this parameter is much worse – on the level of a couple percent. The saturation intensities - $I_s = 6.7$ W/cm² and $I_s = 2.9$ W/cm² were measured for both materials, respectively.

Summing up, this result is opening the bright future for the MIR fast devices. I fully recommend it for publications. It will be a seed paper for many new ideas and new investigations.

Answer

We thank a lot the referee for recognizing the novelty of our manuscript and provided inspiring routes of new ideas and new investigations. We are delighted for this very positive evaluation.

Reviewers' Comments:

Reviewer #1:

Remarks to the Author:

Although the authors haven't gone too far in addressing my comments, it is still a good paper. I am happy to recommend publication.

Reviewer #2:

Remarks to the Author:

Thank you for taking into account my remarks.